# Neural population dynamics and temporal context cells in macaque medial parietal cortex support temporal order memory

Shuzhen Zuo[1,2,3], Chenyu Wang[4,5], Lei Wang[1,6], Zhiyong Jin[1,4], Xufeng Zhou[1,4], Ning Su[1,4], Jianhua Liu[1,4], Thomas J. McHugh[2,3], Makoto Kusunoki[7,8], Sze Chai Kwok [1,4]*

1 Shanghai Key Laboratory of Brain Functional Genomics, Key Laboratory of Brain Functional Genomics (Ministry of Education), School of Psychology and Cognitive Science, East China Normal University, Shanghai, China, 2 Department of Life Sciences, Graduate School of Arts and Sciences, The University of Tokyo, Tokyo, Japan, 3 Laboratory for Circuit and Behavioral Physiology, RIKEN Center for Brain Science, Wako-shi, Saitama, Japan, 4 Phylo-Cognition Laboratory, Duke Kunshan University - The First People's Hospital of Kunshan Joint Brain Sciences Laboratory, Division of Natural and Applied Sciences, Digital Innovation Research Center, Duke Institute for Brain Sciences, Duke Kunshan University, Kunshan, Jiangsu, China, 5 Department of Psychological and Brain Sciences, Boston University, Boston, Massachusetts, United States of America, 6 Institute for Translational Neuroscience of Nantong First People's Hospital, Affiliated Hospital of Southeast University, Jiangsu, China, 7 MRC Cognition and Brain Sciences Unit, University of Cambridge, United Kingdom, 8 Department of Experimental Psychology, University of Oxford, Oxford, United Kingdom

* sze-chai.kwok@st-hughs.oxon.org

## Abstract

Episodic memory involves encoding and remembering the order of events experienced over time. Previous work examining the mechanisms of temporal order memories has focused primarily on the hippocampus and prefrontal cortices, with comparatively less attention paid to population-level memory signals in the medial posterior parietal cortex (mPPC). Combining in vivo multi-unit electrophysiology and a temporal order judgment task with naturalistic cinematic material in macaques, we show that population activity in mPPC exhibits temporally structured dynamics during both encoding and retrieval. During encoding, mPPC neuronal ensembles exhibit gradually evolving activity patterns consistent with temporal context representations embedded in the unfolding video episodes, whereas during retrieval these neurons engage in coordinated, synchronous activity preceding memory-guided decisions. Moreover, trial-by-trial similarity between population activity patterns during encoding and retrieval predicts temporal order judgment performance. A separate control experiment further ruled out eye saccades, fixation patterns, and scan paths as confounding factors contributing to the observed neural dynamics. Together, these findings suggest that mPPC contributes to temporal order memory through population-level representations that integrate temporally extended experience with retrieval-related decision processes, rather than through simple sensory-driven or motor-related responses.

**Data availability statement:** Analysis code and processed data supporting the conclusions of this study are deposited on Zenodo, see https://doi.org/10.5281/zenodo.19280257.

**Funding:** This work received support from the Kunshan Municipal Government research funding grant (24KKSGR017) and Duke University Provost Fund for Duke–DKU Collaborations (https://www.dukekunshan.edu.cn/) (25KINTL013) (SCK). The sponsors or funders did not play any role in the study design, data collection and analysis, decision to publish, or preparation of the manuscript.

**Competing interests:** The authors have declared that no competing interests exist.

**Abbreviations:** AP, anteroposterior; FDR, false discovery rate; GLM, generalized linear model; ITI, inter-trial interval; LDA, linear discriminant analysis; mPPC, medial posterior parietal cortex; MTL, medial temporal lobe; RDMs, representational dissimilarity matrices; RSA, representational similarity analysis; RT, response time; SNRs, signal-to-noise ratios; TOJ, temporal order judgment.

## Introduction

Remembering the temporal order of experiences is a fundamental feature of episodic memory [1–4]. Ample research has implicated the medial temporal lobe (MTL) in different aspects of temporal memory, with evidence from a large number of studies suggesting that neural activities in the MTL, especially the hippocampus, mediate the formation of temporal order memories [5–10], potentially via sequence generation [11]. While temporal information about past events may be present within the hippocampus [12–14], converging evidence suggests that temporal memory and order computations also recruit distributed cortical networks, including the lateral PFC [15,16] and/or the posterior parietal cortex [17,18]. Given the precuneus' role in temporal context [19], time orientation [20], as well as coding for naturalistic content-temporal details [21] and forming new memories [22,23], the medial posterior parietal cortex (mPPC) is well positioned to contribute to the retrospective retrieval of temporal order information.

Theorists posit that successful memory of temporal order requires the brain to link temporal contextual information across encoding and retrieval phases. According to the temporal context model [24–26], when multiple events occur close together in time, they generate overlapping population-level representations whose similarity depends on their temporal proximity. These evolving representations provide a readout of recent events and their temporal history. Empirical evidence on studies of time cells and temporal context cells, which encode the passage of time and organize incoming information [27,28], corroborates key predictions of these theories during the encoding experiences. However, while such temporal coding mechanisms offer a plausible means by which elapsed time may be represented, it remains unclear whether and how temporal context signals established during encoding are later reinstated and utilized to support temporal order judgments at retrieval.

To address these questions, we recorded multi-unit neuronal activity from the monkey mPPC (the precuneus) while animals performed a temporal order judgment task involving still frames extracted from previously viewed videos. We found that population activity in mPPC exhibits temporally structured dynamics during encoding, including gradual representational drift and a spectrum of temporal time constants. During the subsequent memory retrieval, population-level synchrony tracks the accumulation of mnemonic evidence and predicts temporal order judgment accuracy. Moreover, trial-by-trial similarity between encoding and retrieval population activity predicts behavioral performance, consistent with reinstatement of temporally structured representations. Finally, we show that significant temporal order judgment (TOJ)-related modulation remains after accounting for early sensory and motor responses, and that oculomotor behavior cannot explain the observed neural effects. Together, these findings implicate population-level dynamics in mPPC in linking temporally extended experience during encoding with decision-related processes during retrieval.

## Results

In total, we completed 42 daily recording sessions from Monkey Jupiter, 21 sessions from Monkey Mercury (main experiment), and 21 sessions from Monkey Mars

(Experiment 2). Across all animals, we recorded 874 single neurons. In the main experiment, recordings were obtained from 676 neurons across two monkeys (401 neurons from Jupiter and 275 neurons from Mercury). In Experiment 2, 198 neurons were recorded from Monkey Mars. The mean number of simultaneously recorded neurons per session was $9.55 \pm 2.28$ for Jupiter, $13.10 \pm 4.27$ for Mercury, and $9.43 \pm 2.01$ for Mars (mean $\pm$ SD) (S1 Fig).

## Behavioral performance during temporal order judgment

In the main experiment, after watching an 8-s naturalistic video, monkeys performed a TOJ task in which they selected the frame that appeared earlier in the video. Probe frames were drawn from fixed temporal positions to equate temporal distance and temporal similarity across conditions (see Methods), ensuring that neural effects reflect temporal order rather than variability in temporal separation or temporal similarity. Accordingly, temporal similarity was controlled across conditions, yielding two TOJ conditions: an *immediate* condition, in which probe frames were drawn from the first half of the video (5th versus 90th frames), and a *delayed* condition, in which probe frames were drawn from the latter half (95th versus 180th frames; Fig 1A). Behavioral performance in both monkeys was significantly above chance and remained stable across sessions for both conditions (Jupiter: immediate = 66.6%, delayed = 69.8%, overall mean = 67.7%, $P < 0.001$; Mercury: immediate = 62.7%, delayed = 69.8%, overall mean = 64.2%, $P < 0.001$; Fig 1D). In Experiment 2, Monkey Mars also demonstrated robust TOJ performance (accuracy: immediate = 79.07%, delayed = 77.35%; overall mean = 78.21%, against chance $P < 0.001$).

To examine whether temporal-position influenced response speed, we compared reaction times for correct trials in which monkeys selected frames for immediate (Frame 5) versus delayed (Frame 95) conditions. Reaction times were significantly longer for late than for early frame selections on pooled data from both animals (linear mixed-effects model, $P < 0.001$), although there is inconsistency across animals (Jupiter: $P < 0.001$; Mercury: $P = 0.417$; Fig 1E). These results indicate that slower responses in the delayed condition reflect temporal-position–dependent decision processes consistent with forward memory search [29].

## mPPC neurons mediate temporal order judgment processes

We next examined whether neuronal activity in the mPPC is engaged during temporal order judgment. Given prior evidence implicating this region in memory recollection and decision processes [30], we tested whether mPPC neurons exhibit task-related modulation during the TOJ period. To this end, we fit a Poisson generalized linear model (GLM) to the activity of each neuron, including regressors for within session block number, reaction time, response outcome (correct versus incorrect), task condition (immediate versus delayed), response side (left versus right), and binary indicators for distinct trial epochs (pretrial fixation, encoding, delay, TOJ, feedback, and inter-trial interval; see Methods). Neurons that exhibited statistically significant modulation during the TOJ epoch, after accounting for all other task-related and behavioral variables, were classified as *TOJ cells*. 461 of 676 neurons met this criterion (Jupiter: 278; Mercury: 183), with the remaining neurons classified as non-TOJ cells (Jupiter: 123; Mercury: 92) (Fig 2A).

To address the concern that TOJ-related responses may be driven by visual stimulation over and above memory retrieval, we additionally ran a control GLM that explicitly models and removes variance associated with visual-onset responses, allowing retrieval- and decision-related modulation to be assessed more conservatively. We extended the original GLM by decomposing the TOJ epoch into an early visual-onset window (0–200 ms post-probe onset), a post-onset retrieval/comparison window (200 ms post-onset until 200 ms before response), and a response-locked motor window (−200 ms to response), allowing TOJ-related modulation to be assessed after accounting for early sensory and motor-related responses. We found that significant TOJ-decision modulation remained after accounting for early sensory responses as well as putative motor responses (Fig 2B). This supports the interpretation that TOJ-related responses are not entirely reducible to simple sensory responses. Moreover, while pseudo-$R^2$ values are expected to be small in Poisson GLMs applied to single-unit spiking data, TOJ cells did exhibit significantly greater TOJ-related modulation than non-TOJ

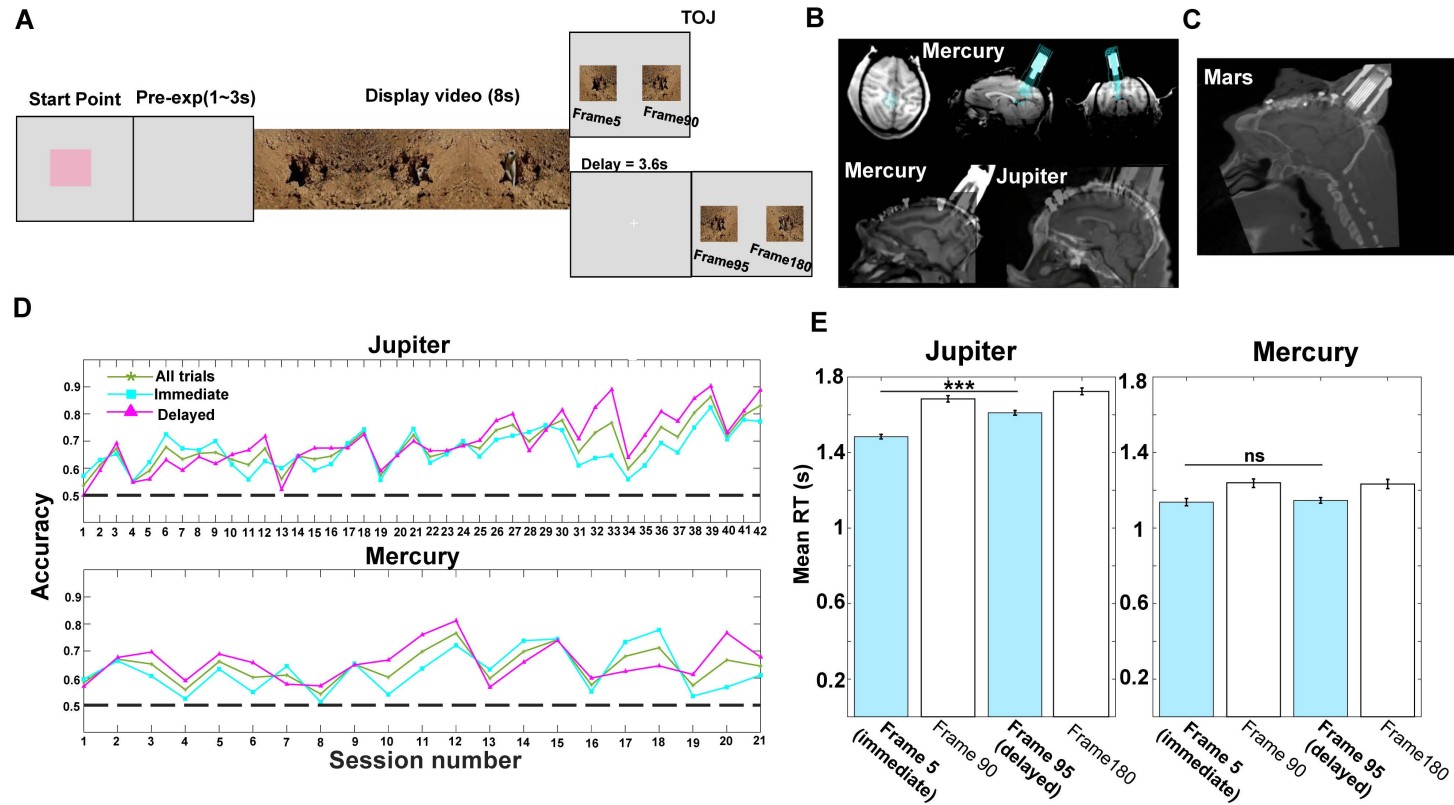

**Fig 1. Paradigm, recording sites, and behavioral performance. A.** Temporal order judgment (TOJ) task. In each trial, the monkey watched an 8-s naturalistic video, and following either a 0-s (immediate) or 3.6-s (delayed) retention interval, judged which of two probe frames had appeared earlier in the video to obtain a reward. All still frames displayed in Fig 1A were generated and assembled by the authors for illustrative purposes of the task design and do not contain any third-party copyrighted material. **B.** Upper panel: Anatomical MRI images of monkey Mercury showing the position of the recording chamber (blue) implanted in the medial posterior parietal cortex (mPPC). Bottom panel: CT images aligned with T1-weighted MRI images visualizing electrode locations in monkeys Mercury and Jupiter. The images were generated by the authors using 3D Slicer software, and no third-party copyrighted material is included. **C.** CT image aligned with T1-weighted MRI showing electrode locations in monkey Mars (Experiment 2). The images were generated by the authors using 3D Slicer software, and no third-party copyrighted material is included. **D.** Behavioral accuracy across all sessions for all trials (green star), immediate condition (cyan square), and delayed condition (magenta triangle) for monkeys Jupiter and Mercury. Data underlying this figure are available in S1 Data. **E.** Reaction times (RTs) as a function of chosen frame location. RTs are shown for trials in which monkeys selected frames from early (Frame 5) versus late (Frame 95) positions within the video. Data are plotted separately for each monkey (Jupiter and Mercury) and include both immediate (0-s delay) and delayed (3.6-s delay) conditions. For the primary analysis, statistical comparisons were performed on correct trials only using linear mixed-effects models, with chosen frame position (early versus late) as a fixed effect and session as a random effect. RTs were significantly longer for late compared with early frame selections ($P < 0.001$). Error trials are shown for completeness but were not included in this primary statistical comparison. Data underlying this figure are available in S1 Data.

cells (Fig 2C). Pseudo-$R^2$ values are significantly larger for the TOJ cells than for non-TOJ cells (Jupiter: two-sample $t$ tests, t (399) = 7.806, $P < 0.001$, Cohen's $d = 0.793$; Mercury: t (273) =16.640, $P < 0.001$, Cohen's $d = 2.029$).

## TOJ neurons exhibit decision-related firing dynamics rather than outcome or firing rate changes

To determine whether TOJ neurons are specifically involved in temporal order judgment rather than reflecting stimulus onset, arousal, or motor preparation, we examined the relationship between neuronal activity and behavioral response time (RT), a continuous index of decision formation. For each neuron, we quantified the trial-wise relationship between firing rate during the TOJ period and RT separately for correct and incorrect trials. These slopes were entered into a linear mixed-effects model with neuron category (TOJ versus non-TOJ), trial outcome (correct versus

**Fig 2. Identification and properties of TOJ-modulated neurons in mPPC.** Data underlying this figure is available in S1 Data. **A.** Stepwise Poisson generalized linear model (GLM) coefficients for task-related variables. Predictors included block number within session, reaction time, trial outcome

(correct versus incorrect), task condition (immediate versus delayed), response side (left versus right), and event-period regressors for pretrial fixation, encoding, delay, TOJ period, feedback (reward versus blank), and inter-trial interval (ITI). TOJ-related regressors exhibited the largest coefficient magnitudes relative to other task variables. Bars indicate mean regression coefficients (±SEM) from GLMs fitted to TOJ neurons, pooled across monkeys. Numerals in brackets indicate the number of neurons significantly modulated by each variable. **B.** Same as (A), except the TOJ period was decomposed into an early visual-onset window (0–200 ms after probe onset), a post-onset retrieval/comparison window (200 ms after probe onset until 200 ms before response), and a response-locked motor window (−200 ms to response), allowing TOJ-related modulation to be assessed after accounting for sensory- and motor-related activity. **C.** Histogram of pseudo-$R^2$ values from Poisson GLM for TOJ cell (cyan) and non-TOJ cell (brown) across two monkeys. Neurons were classified as TOJ cells based on the statistical significance of TOJ-related regressors (A and B), not on pseudo-$R^2$ magnitude. **D.** Trial-wise slopes of the relationship between firing rate during the TOJ period and reaction time, shown separately for TOJ versus non-TOJ neurons and for correct versus incorrect trials. **E.** Example neurons showing higher firing rates on correct than incorrect trials during the TOJ period. Left panels show spike rasters for correct (top) and incorrect (bottom) trials. Right panels show mean firing rates for correct (black) and incorrect (red) trials for the example neuron (top) and the population average (bottom). **F.** Same as (E), but for an example neuron showing higher firing rates for incorrect than correct trials.

incorrect), and their interaction as fixed effects, and animal identity as a random effect. This analysis revealed a robust main effect of neuron category: TOJ neurons exhibited a significantly stronger negative RT–firing rate relationship than non-TOJ neurons (mean difference = −0.593, $P = 3.29 \times 10^{-7}$), indicating that higher firing rates in TOJ neurons were associated with faster temporal order judgments. In contrast, neither the main effect of trial outcome nor the interaction between neuron category and trial outcome was significant, indicating that this coupling between neuronal activity and decision speed was present in both correct and incorrect trials (Fig 2D). Thus, TOJ neuron activity tracks the process of temporal order evaluation itself rather than outcome monitoring or post-decisional signals. Consistent with this interpretation, overall firing rates during the TOJ period did not differ between correct and incorrect trials for either TOJ or non-TOJ neurons. To assess whether TOJ-related activity reflected outcome-dependent firing rate differences, we compared firing rates during the TOJ period between correct and incorrect trials. Using two-tailed tests, a small subset of neurons (59/676, 8.7%) showed significant differences (Jupiter: 23/401 neurons; Mercury: 36/275 neurons; all two-tailed $Ps < 0.05$; Fig 2E, 2F). Importantly, ~45% of these neurons belonged to the TOJ category, consistent with the absence of significant interaction between neuron category and trial outcome in the population-level model.

## Spike-train synchrony in mPPC tracks evidence accumulation during temporal order judgment

To assess population-level coordination during TOJ, we quantified spike-train synchrony using the SPIKE-distance metric, a time-resolved measure of relative spike timing normalized to local firing rates (lower values indicate higher synchrony). SPIKE-distance was computed across all neuron pairs within each session and averaged to obtain a population synchrony time course, aligned either to TOJ onset (probe presentation) or to TOJ response offset (manual decision). Following TOJ onset, synchrony increased rapidly and peaked within ~200 ms (Fig 3A, 3B, top), consistent with a transient, stimulus-locked coordination of neural activity. No reliable differences between correct and incorrect trials were observed during this early period, suggesting that this initial synchrony reflects shared sensory input or general task engagement rather than decision-specific processing. In contrast, when synchrony was aligned to response time, a clear correctness-related effect emerged. Population synchrony was significantly higher on correct than incorrect trials during the period preceding the decision (Fig 3A, 3B, bottom), with differences emerging ~500–1,000 ms before response onset. This gradual divergence indicates that coordinated spike timing across mPPC neurons tracks the accumulation of mnemonic evidence leading to successful temporal order judgments. Moreover, synchrony also differed across task conditions. Synchrony was higher in the immediate than in the delayed condition when aligned to TOJ onset (Fig 3C, 3D, top), but this difference was largely absent when aligned to response time (Fig 3C, 3D, bottom). This dissociation suggests that across-condition synchrony differences primarily reflect task-state factors such as shared sensory drive and retention demands at probe onset, rather than decision-related processing per se. Accordingly, higher global synchrony does not necessarily predict better behavioral performance. Rather, spike synchrony reflects the underlying network state and

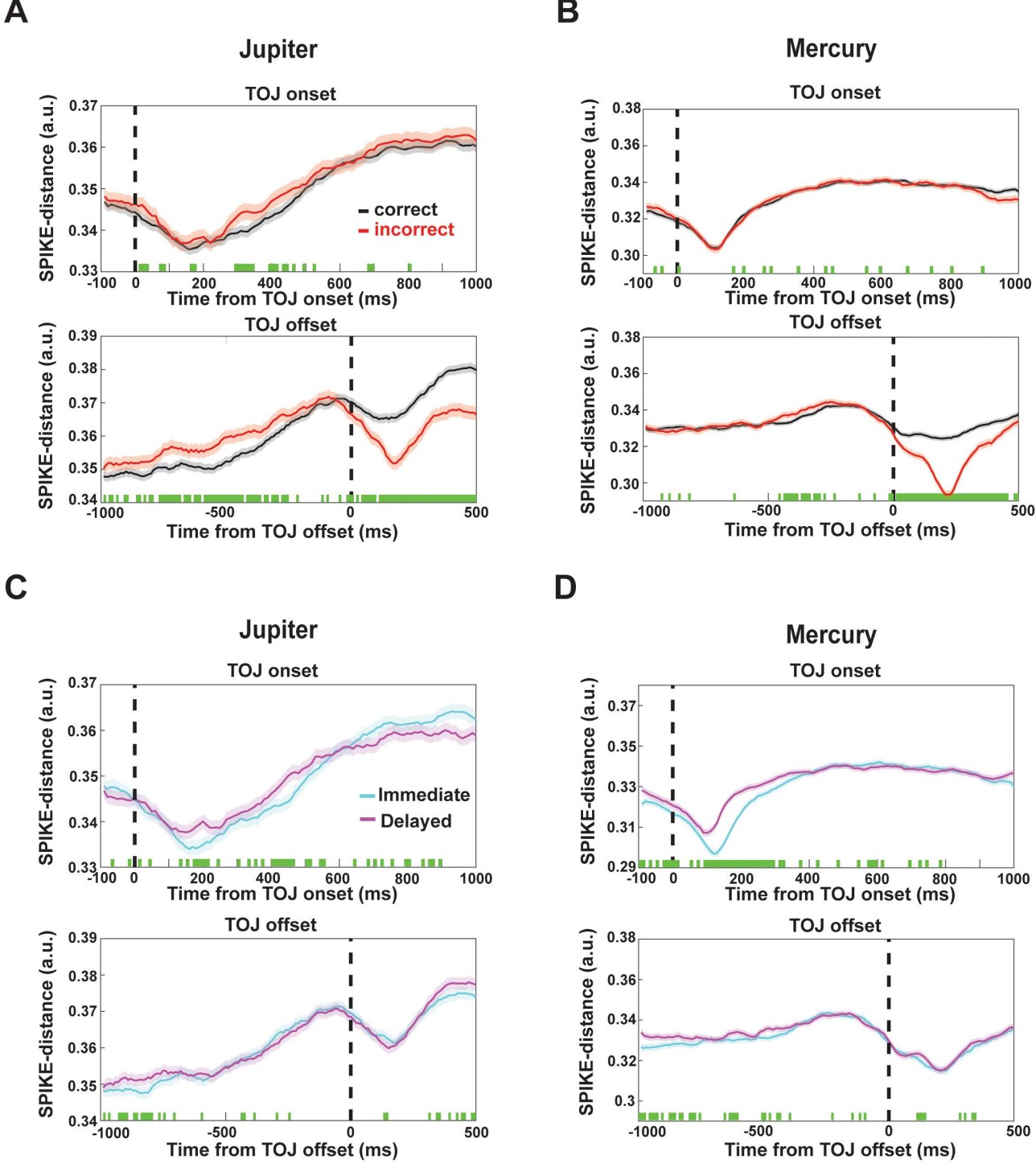

**Fig 3. Population spike-train synchrony tracks temporal order retrieval.** Data underlying this figure are available in S1 Data. **(A)** and **(B)** Population spike-train synchrony quantified using SPIKE-distance. Upper panels show synchrony time-locked to TOJ probe onset (0 ms). Lower panels show synchrony time-locked to the behavioral response (−1,000 to +500 ms). Black traces indicate correct trials and red traces incorrect trials. Green dots mark time bins showing significant differences between correct and incorrect trials (paired *t* tests, *P* < 0.01). Shaded regions denote SEM across sessions. We also noted a divergence in SPIKE-distance after the response, which might be attributed to post-decision, non-mnemonic processes. **(C** and **D)** Comparison of spike-train synchrony between immediate and delayed conditions for the two monkeys. Synchrony was consistently higher in the immediate

condition when aligned to probe onset, whereas condition differences were reduced when aligned to response time, suggesting task-state-dependent modulation. Legends are the same as in (A and B).

coding regime [31], whereas successful temporal order judgments depend on the stability and fidelity of temporal context representations.

Although stimulus onset can induce transient, event-locked changes in neural activity, several lines of evidence argue against a purely stimulus-driven or firing rate–driven explanation for the observed synchrony effects. First, synchrony dynamics differed across task conditions (immediate versus delayed), despite identical probe stimuli, indicating modulation beyond a simple onset-related reset. Second, the temporal profiles of synchrony for correct versus error trials differed between TOJ onset and the period preceding the behavioral response, with correctness-related effects emerging primarily before decision execution rather than immediately after stimulus presentation. Third, population firing rates per se did not differ between correct and incorrect trials, and the subset of neurons showing outcome-related firing rate modulation was sparse and distributed across sessions. Critically, synchrony was quantified using the SPIKE-distance metric, which measures relative spike timing independently of firing rate. Together, these observations indicate that enhanced synchrony during correct trials reflects coordinated temporal organization of neuronal activity associated with retrieval and decision processes, rather than trivial differences in firing rate or stimulus-evoked responses.

The divergence between correct and incorrect trials was most pronounced after TOJ offset. This post-offset effect was observed in the correctness contrast but not in the immediate versus delayed comparison (Fig 3C, 3D), suggesting that it reflects processes downstream of temporal order computation. Neural activity following TOJ offset likely includes post-decisional processes such as commitment to a choice, outcome evaluation, and consummatory signals, as correct trials were consistently followed by reward delivery. In contrast, the synchrony difference observed during the TOJ period occurred prior to response execution and reward, temporally aligning with the period in which temporal order information is actively evaluated. Although smaller in magnitude, this during TOJ effect therefore provides more direct evidence for mnemonic and decision-related processing underlying temporal order judgment.

## Identification of temporal context cells during encoding period

Temporal context cells were identified based on their transient responses to the onset of the video, followed by exponentially decaying activity with heterogeneous time constants. Following the framework established in prior work on temporal context representations, the onset of a salient event serves as a temporal anchor, and elapsed time since that event is encoded implicitly in the distributed population state formed by neurons with different relaxation times [27] via Laplace transform [28].

We examined whether neurons in the mPPC exhibit activity consistent with temporal context representations during memory encoding. Following prior work in the primate entorhinal cortex, we sought to identify temporal context cells—neurons whose activity is transiently modulated at episode onset and decays over time with heterogeneous time constants, thereby contributing to a population-level representation of elapsed time [27, 32]. To this end, we fitted each neuron's firing activity during video viewing with three nested models (see *Temporal context cell fitting* in Methods): a constant model, a Gaussian model, and an ex-Gaussian model consisting of a Gaussian convolved with an exponential decay. The ex-Gaussian model is characterized by two key parameters: the Gaussian mean ($\mu$), which captures response latency relative to video onset, and the exponential time constant ($\tau$), which reflects how long the neuron takes to relax back toward baseline (63% decay from peak firing). Model comparison using likelihood ratio tests identified 111 of 676 neurons as temporal context cells. Among these neurons, 63 showed increased firing rates at video onset (Fig 4A, left), whereas 48 showed decreased firing rates (Fig 4A, right). Notably, a subset of neurons displayed particularly long decay profiles, resembling temporal context cells previously reported in the primate entorhinal

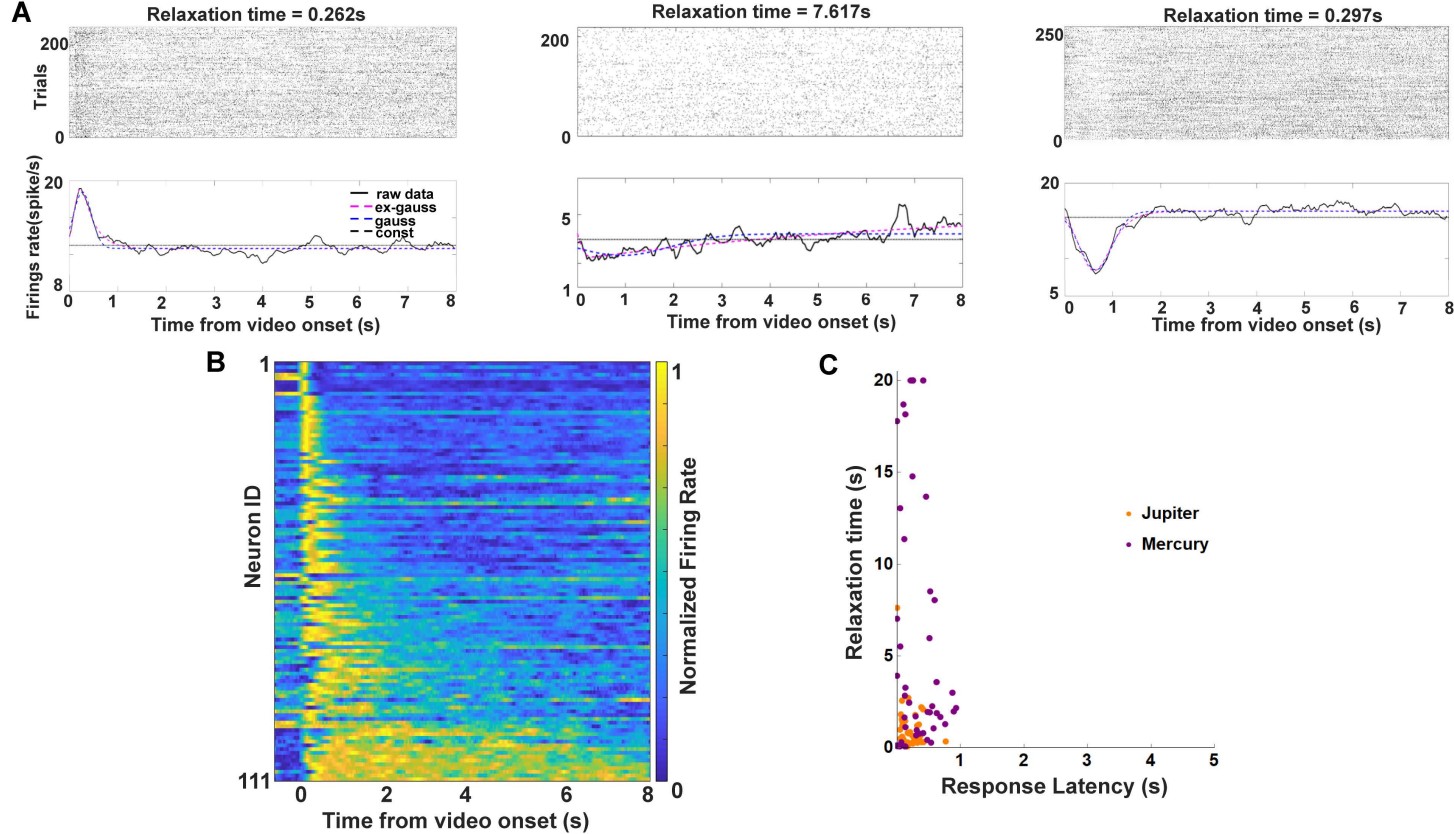

**Fig 4. Temporal context cells during video encoding.** Data underlying this figure is available in S1 Data. **A.** Three example temporal context cells recorded during video viewing. Spike raster (top) and peristimulus time histograms (PSTHs; bottom) are aligned to video onset. Solid black lines indicate smoothed firing rates. Dashed pink, blue, and black lines indicate fits of the ex-Gaussian, Gaussian, and constant models, respectively. Relaxation time is defined as the interval between peak firing and 63% return to baseline. Left and right examples show transient increases or decreases at video onset, whereas the middle example shows a prolonged decay characteristic of temporal context cells. **B.** Heat map of normalized firing rates of 111 temporal context cells aligned to video onset and sorted by relaxation time, illustrating a broad spectrum of decay rates spanning the video duration. **C.** Scatter plot of response latency versus relaxation time for temporal context cells. Response latencies were <1 s for all neurons, whereas relaxation times ranged from near zero to ~20 s. Colors indicate neurons recorded from monkey Jupiter (orange; $n = 66$) and Mercury (purple; $n = 45$).

cortex [27] (Fig 4A, middle). At the population-level, normalized firing rates revealed that most temporal context cells responded shortly after video onset. These neurons exhibited a broad distribution of relaxation times (Fig 4B). While response latencies clustered within the first second following video onset, decay time constants spanned a wide range, extending from near zero to several seconds to a hypothetical 20 s (Fig 4C). Together, this heterogeneity in decay dynamics suggests that mPPC temporal context cells collectively span multiple timescales. Importantly, we do not assume that individual neurons explicitly label specific moments or frames within the video. Rather, temporal information is represented at the population-level, such that these cells would have different relaxation times and be relaxing to the baseline at different rates to form a whole course temporal memory of the videos. The discriminability of temporal order arises from population-level representational geometry, not from frame-specific or event-specific coding by individual neurons. While temporal context cells are defined at the single-neuron level based on their encoding dynamics, these neurons represent only one component of the broader population activity. To assess how temporally structured information is represented at the population-level, we next examined whether ensemble activity across mPPC encodes elapsed time during video viewing.

## Cell ensembles carry information about passage of time during encoding and predict subsequent memory performance

We next asked whether population activity in the mPPC carries information about the passage of time during video encoding. Importantly, this population-level decoding analysis does not rely exclusively on identified temporal context cells but instead reflects distributed activity across the recorded neuronal population, allowing us to assess how temporal information is represented at the ensemble-level. To address this, we trained a linear discriminant analysis (LDA) decoder to estimate elapsed time from ensemble neural activity. Reliable prediction of the correct time bin for held-out data would indicate that population responses encode temporal progression. For each trial, the 8-s encoding period was divided into 32 nonoverlapping time bins of 250 ms. Within each session, population activity in each bin was represented as a vector of firing rates across simultaneously recorded neurons. Temporal decoding was performed using LDA with a standardized 5-fold cross-validation procedure (see Methods), ensuring independence between training and test data. Decoding performance was quantified as the mean absolute temporal error between the true and decoded time bins (Fig 5A and 5D) and assessed against a null distribution generated by permuting time labels (1,000 iterations). Across sessions, decoding errors were significantly lower than chance. Mean absolute decoding errors were 1.44 s for Jupiter and 1.50 s for Mercury, compared with chance-level errors of $2.66 \pm 0.18$ s and $2.66 \pm 0.20$ s, respectively (Fig 5B and 5E). Although decoding accuracy was highest during the earliest portion of the video (Fig 5C and 5F), robust above chance decoding persisted after progressively excluding early time bins in 0.5-s increments (S2 Fig), indicating that temporal information was distributed throughout the encoding period rather than confined to stimulus onset.

To characterize how temporal context evolves over time, we performed a representational similarity analysis (RSA) on population activity during encoding. Neural activity was binned into 250-ms windows, z-scored across neurons, and pairwise Pearson correlations were computed between all of the time bins to construct representational similarity matrices. Population similarity was highest between temporally adjacent bins and decayed monotonically with increasing temporal separation (Fig 5G), quantifying the rate at which neural representations drifted over time during movie watching. This decay profile quantifies the rate at which population-level neural representations drift over time during encoding, consistent with a gradually evolving temporal context signal. Moreover, to assess behavioral relevance with neural representations, we examined whether session-level decoding accuracy predicted subsequent temporal order judgment performance. Across recording sessions ($n = 63$), LDA decoding accuracy during encoding was positively correlated with behavioral accuracy during retrieval ($r = 0.26$, $P = 0.039$; Fig 5H), indicating that session-to-session variability in ensemble-level neural discriminability during encoding predicts subsequent TOJ performance during retrieval. Together, these analyses show that mPPC neuronal ensembles encode elapsed time during encoding, that population activity evolves gradually across the episode rather than reflecting only transient sensory responses, and that the strength of this temporal representation predicts subsequent TOJ performance.

### Encoding–retrieval neural similarity supports temporal order judgment

To further link these population-level representations to memory retrieval, we examined the similarity between ensemble activity patterns during encoding and the TOJ period. To assess whether population-level neural activity in mPPC links encoding and retrieval during temporal order judgment [33,34], we quantified the similarity between ensemble firing patterns during video viewing and TOJ period. As with the decoding analysis, this approach captures distributed population dynamics rather than being restricted to identified temporal context or TOJ cell categories. For each trial, we constructed N-dimensional population vectors from firing rates during encoding and retrieval, where N denotes the number of simultaneously recorded neurons in that session. To allow comparisons across sessions with different neuron counts, distances were normalized by twice the number of neurons in each session, yielding a Mahalanobis distance index [6]. We then used the Mahalanobis distance between these vectors as a measure of dissimilarity between encoding and retrieval population states (Fig 6A). Across monkeys, population activity

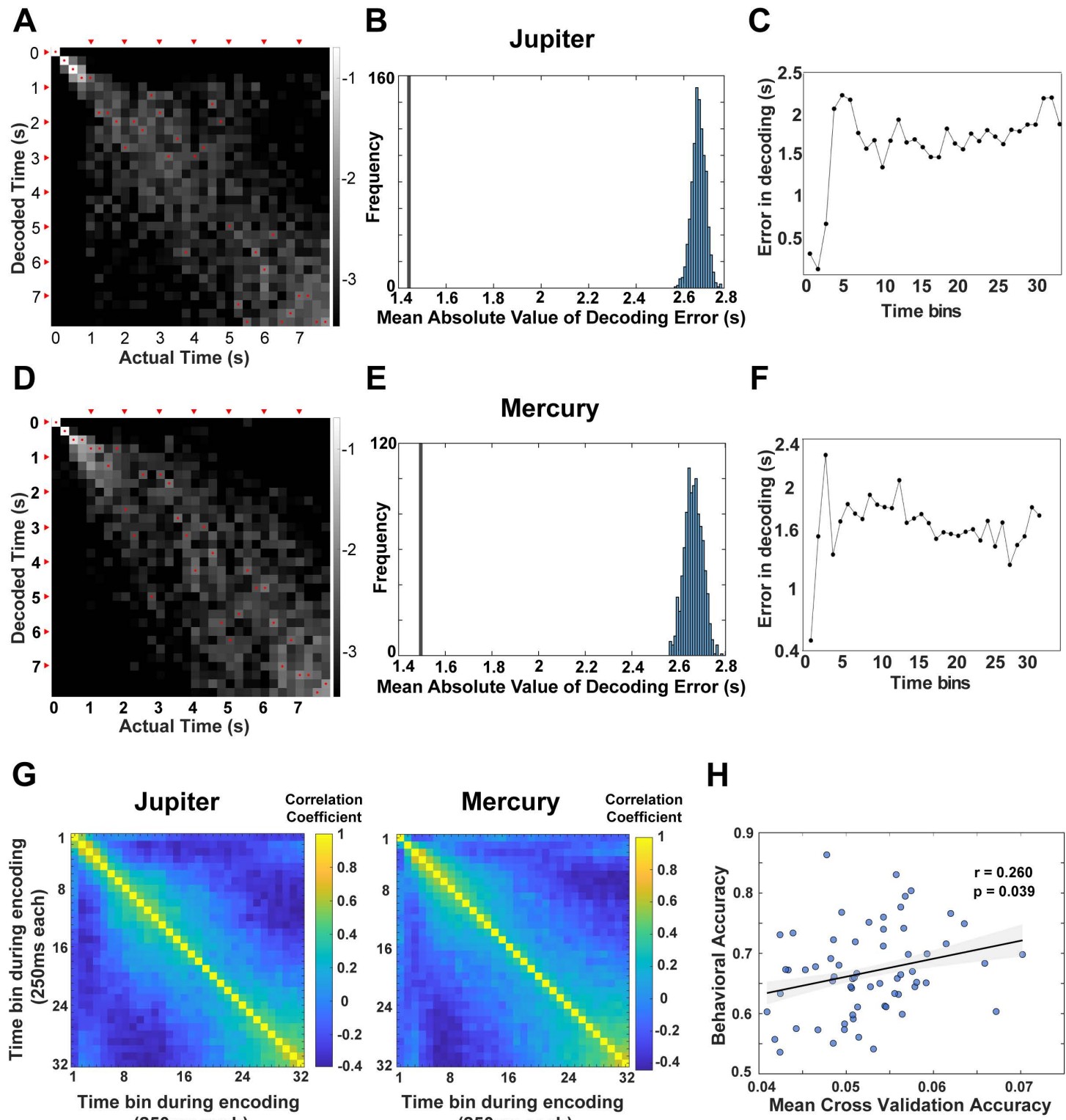

**Fig 5. Population decoding of elapsed time during video encoding.** Data underlying this figure is available in S1 Data. **A.** Temporal decoding perfor-
mance based on population activity of mPPC neurons in monkey Jupiter. The x-axis indicates actual time bins and the y-axis decoded time bins (250-ms

resolution; 32 bins per video). Grayscale denotes posterior probability (log scale), with lighter values indicating higher probability. Red dots mark the maximum-probability decoded time for each bin; red arrows denote 1-s intervals. **B.** Mean absolute decoding error (vertical line; 1.44 s for Jupiter, 1.50 s for Mercury) compared with a permutation-based null distribution. Observed decoding errors were significantly smaller than chance for both monkeys. **C.** Absolute decoding error as a function of time within the video, showing lower error early in the encoding period (see also S2 Fig). **(D)–(F).** Same analyses as (A–C) for monkey Mercury. **(G)** Representational dissimilarity matrices showing pairwise correlations between population activity patterns across all time bins during encoding for each monkey. **(H)** Relationship between session-level decoding accuracy during encoding and subsequent TOJ performance. Each point represents a session ($n = 63$), collapsed across monkeys.

patterns during TOJ were significantly more similar to encoding patterns on correct than on incorrect trials (Fig 6B). Specifically, distance indices were significantly lower for correct trials than for incorrect trials (Jupiter: correct mean = 0.517 ± 0.726 SD, incorrect mean = 0.664 ± 0.701 SD; Mercury: correct mean = 0.913 ± 0.543 SD, incorrect mean = 1.086 ± 0.821 SD; both $Ps < 0.001$), indicating greater encoding–retrieval similarity during successful temporal order judgments. To determine whether these similarity effects exceeded chance levels, we conducted a permutation analysis in which firing rate matrices for encoding and TOJ periods were independently shuffled 1,000 times within session, thereby preserving marginal firing statistics while disrupting trial-specific correspondence. Mahalanobis distance indices observed under task conditions were significantly smaller than those obtained from permuted data for both correct and incorrect trials ($P < 0.001$; Fig 6C), demonstrating that encoding–retrieval similarity was reliably greater than chance.

To rule out the possibility that reduced encoding–retrieval Mahalanobis distance in correct trials was driven by stronger visual responses early during encoding, we explicitly examined whether encoding–retrieval similarity depended on which half of the video contributed to the encoding population vector. Specifically, we computed Mahalanobis distance separately using firing rate vectors derived from the first half (0–4 s) or the second half (4–8 s) of the encoding period and fitted a linear mixed-effects model including trial correctness and encoding video half as predictors. This analysis revealed a robust reduction in Mahalanobis distance for correct relative to incorrect trials ($P < 0.001$), whereas encoding video half had no significant main effect ($P = 0.240$) and did not interact with correctness ($P = 0.230$). Thus, encoding–retrieval similarity was not explained by stronger visual responses early in the video but instead reflected memory-related reinstatement processes. These findings indicate that mPPC population activity exhibits trial-specific reinstatement of encoding patterns during temporal order judgment, and that the fidelity of this reinstatement predicts successful memory performance, consistent with prior evidence for neural pattern reinstatement during episodic recall [35–38].

### Relationship between Temporal Context Cells × TOJ Cells

We next examined how population-level similarity effects relate to temporal context coding. Among the 676 neurons recorded in the main experiment, 111 were classified as temporal context cells and 461 as TOJ cells, with 78 neurons belonging to both categories (Fig 6D). Under an independence assumption, the expected overlap was 75.7 neurons, which did not differ from the observed count ($\chi^2(1) = 0.084$, $P = 0.772$). In addition, the proportion of temporal context cells did not differ between TOJ and non-TOJ neuron populations ($\chi^2(1) = 0.162$, $P = 0.688$). The overlap between temporal context cells and TOJ cells was not greater than expected by chance (S1 Table), indicating that these functional properties are not preferentially co-localized at the single-neuron level but instead contribute to distributed population dynamics. A heat map of normalized firing rates for the 78 overlapping neurons revealed activity spanning the encoding period (Fig 6E), consistent with contributions to distributed temporal representations. However, while some neurons exhibit both properties, temporal context cells are not preferentially recruited during temporal order judgment. Together, these findings suggest that temporal order memory in mPPC does not rely on a specialized subset of dual-function neurons but instead emerges from population-level interactions that integrate temporally structured encoding activity with retrieval-related decision processes.

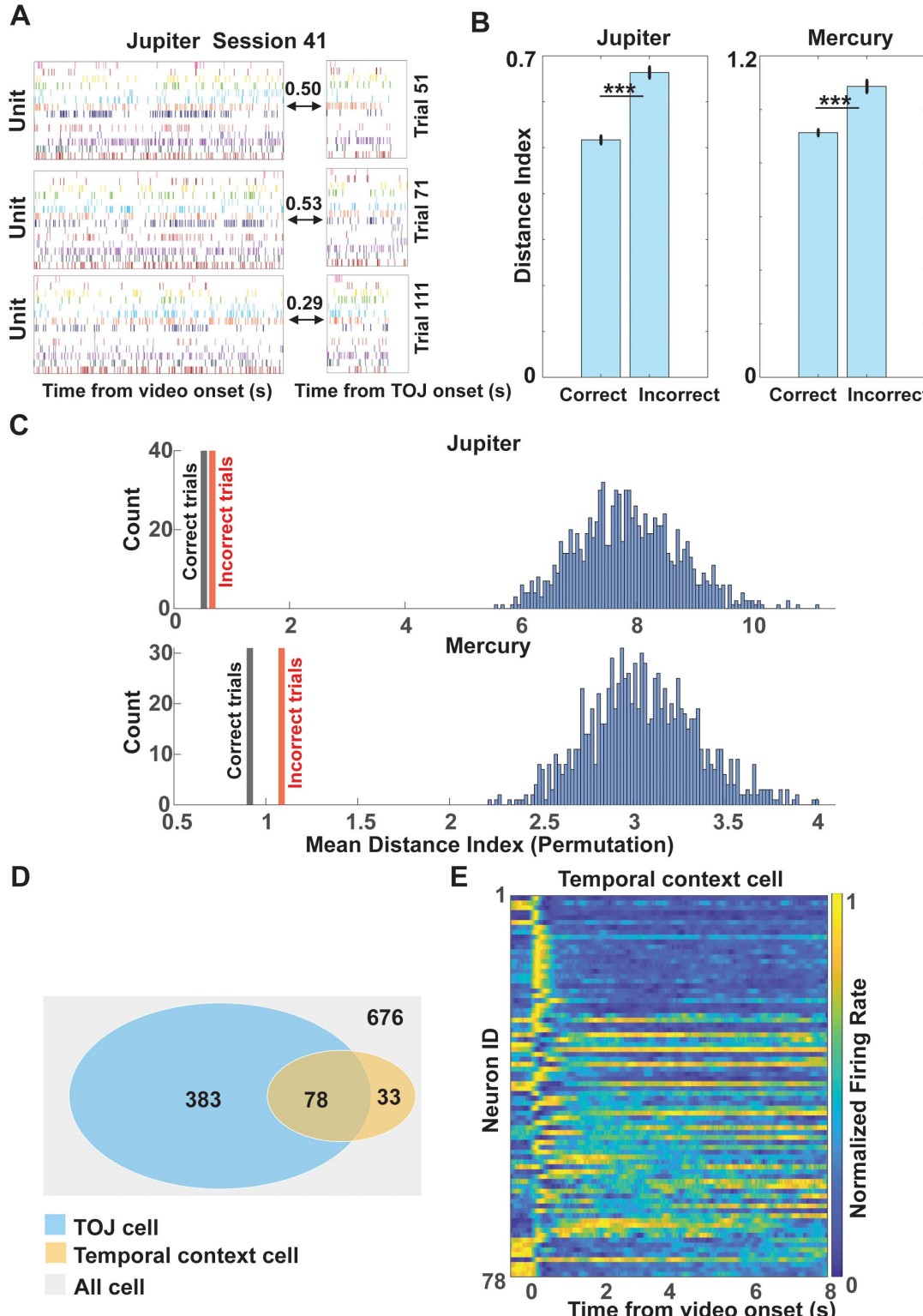

**Fig 6. Encoding–retrieval similarity and overlap between temporal context and TOJ cells.** Data underlying this figure is available in S1 Data. **A.** Example trials illustrating population activity patterns during encoding and the TOJ period within a single session (monkey Jupiter, session 41). Values indicate the normalized Mahalanobis distance (Distance Index) between encoding and retrieval population vectors for each trial. **B.** Comparison of

encoding–retrieval similarity for correct and incorrect trials. Correct trials exhibited significantly smaller distance indices, indicating greater population-level similarity between encoding and retrieval states. * $P<0.05$, ** $P<0.01$, *** $P<0.001$. **C.** Observed Mahalanobis distance indices for correct and incorrect trials compared with permutation-based controls. Distances under task conditions were significantly smaller than permuted values for both monkeys ($P<0.001$). **D.** Venn diagram showing overlap between temporal context cells (111 cells) and TOJ cells (461 cells) among all recorded neurons ($n=676$), with 78 neurons shared between the two categories. **E.** Heat map of normalized firing rates for the 78 neurons overlapping between temporal context and TOJ cell populations.

## Eye-movement confounds are unlikely to account for TOJ-related neuronal activity

To assess whether TOJ-related neuronal activity could be explained by eye movements during the decision period, we conducted a dedicated control experiment in a third monkey (Experiment 2) in which eye position, scan paths, and neuronal activity were recorded simultaneously during task performance (see S1 Video). This design allowed us to directly quantify how neuronal firing varied with overt visual exploration of the TOJ probe images. We first visualized firing rates as a function of fixation locations and saccadic scan paths for representative neurons, separately for correct and incorrect trials (S3A, S3B Fig). These examples confirmed that monkeys actively inspected both probe frames during TOJ but did not reveal a systematic mapping between eye position or scan trajectory and neuronal firing that could trivially explain TOJ-related modulation. To formally control for eye-movement effects, we extended the Poisson GLM to include eye-movement regressors—saccade frequency, fixation frequency, total scan-path length, and total fixation duration—alongside all task-related variables. After accounting for these factors, the number of neurons classified as TOJ cells decreased from 58 to 41. Critically, the majority of these TOJ cells (70.7%, 29/41) did not retain any eye-movement regressor as a significant predictor. Only a small subset showed modulation by fixation duration ($n=6$), fixation frequency ($n=5$), or saccade frequency ($n=1$) (S2 Table). Importantly, a comparable proportion of non-TOJ neurons also showed no significant eye-movement modulation (82.8%, 130/157), indicating that eye-movement variables did not differentially explain TOJ-related versus non-TOJ-related neuronal activity. Together, these results demonstrate that eye movements account for only a minor fraction of TOJ-related firing and do not selectively drive the neuronal signatures associated with temporal order judgment. Finally, we emphasize that Experiment 2 differs from the main experiment in stimulus structure and is not intended as a replication. Rather, it serves specifically to rule out oculomotor confounds, providing convergent evidence that TOJ-related neuronal activity in mPPC cannot be explained by fixation patterns, saccades, or scan-path metrics alone.

## Discussion

In this study, we investigated how neuronal activity in the macaque mPPC supports temporal order memory during both encoding and retrieval of naturalistic events. Our findings reveal a dissociable but complementary set of neural signatures: during encoding, mPPC populations exhibit a slowly evolving temporal context representation spanning a broad range of timescales; during retrieval, population-level coordination and reinstatement-like similarity between encoding and retrieval states track decision formation and behavioral accuracy. Importantly, our analyses emphasize population-level representations and task-evoked dynamics rather than assuming that single neurons explicitly "time-stamp" individual moments. Together, these results position mPPC as an important component of the broader circuit supporting ordinal memory and temporal organization of experience.

### Retrieval-related activity in mPPC reflects decision dynamics rather than sensory or outcome signals

First and foremost, we examined the single and ensemble neuronal activities to understand the mechanisms of mPPC involved in temporal memory retrieval. A substantial proportion of mPPC neurons exhibited modulation during the TOJ period. Importantly, this modulation persisted after accounting for visual-onset responses and motor execution by decomposing the TOJ epoch into sensory-driven, retrieval/comparison, and response-locked components. The persistence of

TOJ-related modulation in the post-onset window indicates that mPPC activity during TOJ cannot be reduced to stimulus-evoked or motor-related responses alone, supporting its involvement in memory-guided decision processes rather than purely perceptual or motor functions [30,38]

Further evidence for a retrieval-related role comes from the relationship between neuronal firing and behavioral response time. TOJ-classified neurons exhibited a stronger negative relationship between firing rate and reaction time than non-TOJ neurons, indicating that higher firing rates were associated with faster temporal order judgments. Critically, this relationship was present in both correct and incorrect trials, suggesting that TOJ neurons track the process of temporal order evaluation itself rather than outcome monitoring, reward expectation, or feedback [17, 18]. This pattern mirrors human neuroimaging findings showing that precuneus activity during temporal order retrieval is engaged independently of accuracy [18,39] pointing to a conserved role of medial parietal cortex in temporal evaluation [38] rather than error signaling across primate species.

At the single-neuron level, differences in firing rate between correct and incorrect trials were modest, sparse, and variable across animals, and they were not preferentially expressed by TOJ neurons. This dissociation underscores an important point: successful temporal order memory is not primarily reflected in outcome-dependent firing rate changes of individual neurons, but rather in population-level dynamics that unfold during retrieval, consistent with prior work emphasizing ensemble-level coding during memory-guided decisions [30,40,41].

## Population synchrony tracks evidence accumulation during temporal order judgment

Consistent with a population-level account, spike-train synchrony revealed dynamics that were not apparent from firing rates alone. Synchrony increased rapidly and peaked around 250 ms following TOJ probe onset, likely reflecting shared sensory input and task engagement, but this early synchrony did not differentiate correct from incorrect trials. In contrast, when activity was aligned to the time of the behavioral response, a clear divergence emerged: synchrony was significantly higher on correct trials than incorrect trials during the period preceding the decision. This gradual divergence is consistent with a role for coordinated population activity in evidence accumulation leading to successful temporal judgments, as observed in decision-making, working memory, and perceptual inference tasks [42–44].

Notably, synchrony differed across task conditions: synchrony was higher in the immediate than in the delayed condition when aligned to probe onset, despite behavioral performance being comparable or better in the delayed condition. This dissociation indicates that higher global synchrony does not necessarily reflect stronger memory. Instead, synchrony appears to index the underlying network state and coding regime—such as shared sensory drive or task engagement—rather than memory strength per se [42]. Within a fixed task regime, however, enhanced synchrony during the decision phase predicts successful judgments, consistent with findings in inferotemporal and parietal cortices showing that coordinated spike timing supports correct perceptual and mnemonic decisions [40,41,45].

Critically, these synchrony effects cannot be explained by trivial firing rate differences. Population firing rates did not differ between correct and incorrect trials, and the synchrony metric used explicitly controls for firing rate by quantifying relative spike timing [46,47]. Moreover, correctness-related synchrony effects emerged late in the trial, whereas stimulus-locked synchrony peaked early, arguing against a simple event-driven reset account. Together, these findings indicate that population coordination—rather than single-neuron firing rate—is the primary neural signature distinguishing successful from unsuccessful temporal order judgments in mPPC.

## Encoding-related temporal context representations in mPPC span multiple timescales

During encoding, we identified a subset of mPPC neurons whose activity was transiently modulated at video onset and decayed back toward baseline with heterogeneous time constants. These temporal context cells resemble those previously reported in mammalian entorhinal cortex and hippocampus [27,32,48,49] and are well described by models in which elapsed time since an event boundary is encoded implicitly in a distributed population state [24–26]. These temporal

context cells, which are characterized by their wide spectrum of the past, are distinguished from neurons that are instantly responsive to hard or soft video boundaries in the human MTL [35] or order-selective cells [50]. With an improved set-up such as an ethogram [21,51], it would be possible to expect certain cells are triggered by a certain feature and would relax back at different time scales [28].

Importantly, we do not propose that individual neurons label specific moments or frames within the video. Rather, temporal information is represented at the population-level through the relative activity of neurons with different relaxation times. Because these neurons relax at different rates, the population state evolves continuously over the course of the episode, providing a graded representation of elapsed time. Such a spectrum-based temporal code aligns with theoretical frameworks proposing that neural populations represent the Laplace transform of the recent past [25–28] and is consistent with hierarchical organization of intrinsic timescales across cortex, with parietal regions exhibiting longer temporal receptive windows [52,53]. This spectrum-based temporal record of the past has previously been observed in hippocampal neurons, which hold incremental timing signal for separate items within an event [54], as well as for temporal space within discrete ordinal positions [50] and code for temporal order relationships between events [55,56].

At the ensemble-level, mPPC populations reliably encoded the passage of time during video viewing. Temporal decoding was robust across sessions, persisted after excluding early encoding bins, and revealed a gradual representational drift over time. Notably, session-level decoding accuracy predicted subsequent TOJ performance, linking the fidelity of encoding-related temporal representations to later memory decisions. These findings converge with prior evidence that cortical regions outside the hippocampus can encode temporally extended structure in naturalistic stimuli [19,23,53,57].

While we emphasize that the ability to decode elapsed time from neuronal activity does not, by itself, constitute evidence for a temporal context representation. As has been noted previously, visually responsive neurons in many cortical areas may carry sufficient stimulus-driven dynamics to allow decoding of time during extended visual sequences, without necessarily supporting temporal order or 'when' memory. In the present study, our use of the term temporal context follows the theoretical framework developed by Howard and colleagues and aligns closely with prior work on associative and contextual coding in the primate medial temporal lobe and connected cortical areas [54,58,59]. Within this framework, temporal context is not defined by time decodability or visual responsiveness per se, but by a slowly evolving population state that links temporally separated events and supports temporal order and associative memory.

Consistent with this distinction, nonhuman primate studies show that strong stimulus-driven responses in ventral visual cortex (e.g., area TE) do not necessarily entail temporal order or 'when' coding, whereas medial temporal lobe structures more directly support item–time integration and temporal order memory [16,54]. Accordingly, we do not interpret our decoding results alone as evidence for temporal context but rather emphasize convergent evidence that population activity in mPPC evolves gradually across an episode, predicts subsequent temporal order judgment performance, and reinstates encoding-related population states during retrieval. Importantly, we do not claim that mPPC uniquely or independently encodes temporal context. Instead, temporal context signals may emerge in cortical regions that integrate sensory information over extended timescales [21] and interact with medial temporal lobe memory systems. In this sense, our findings are intended to complement, rather than replace, existing accounts of temporal order memory grounded in medial temporal lobe circuitry [58,59].

### Encoding–retrieval similarity supports reinstatement-like access to temporal context

Beyond encoding and retrieval dynamics considered separately, we found that population activity during TOJ was more similar to encoding activity on correct than incorrect trials. This encoding–retrieval similarity exceeded chance levels and could not be explained by stronger visual responses early in the video. These findings are consistent with reinstatement-like access to encoding representations during retrieval, a hallmark of episodic memory observed across hippocampal, parahippocampal, and parietal regions [33–37].

Notably, temporal context cells were not preferentially enriched among TOJ neurons, and the overlap between these two classes did not exceed chance. This result suggests that temporal order memory in mPPC does not rely on a specialized subset of dual-function neurons. Instead, reinstatement-like similarity arises from interactions between encoding-related temporal context representations and retrieval-related population dynamics distributed across partially overlapping neuronal ensembles. This distributed architecture is consistent with models in which temporal order is inferred from similarity relationships between population states rather than explicit timestamping of individual events [24–26,35].

### Controls, limitations, and broader implications

A dedicated control experiment incorporating eye-tracking demonstrated that TOJ-related neuronal activity cannot be explained by saccades, fixation patterns, or scan paths. Although this experiment differed from the main task and was not intended as a replication, it effectively ruled out oculomotor confounds as a primary driver of TOJ-related activity, consistent with prior evidence that parietal memory signals persist beyond eye-movement explanations [41,60].

Several limitations remain. While our analyses dissociate sensory, motor, and retrieval-related components, stimulus-specific responses may still contribute to population dynamics. Moreover, causal manipulations will be required to determine whether mPPC activity is necessary for temporal order judgments or reflects interactions with medial temporal lobe and prefrontal circuits known to support temporal memory [16,20,54,60–65]. Future work combining simultaneous recordings across regions or perturbation approaches will be essential to clarify these interactions. It would also be valuable to apply canonical replay/reactivation methodologies to examine neural dynamics during retrieval and post-retrieval periods to elucidate TOJ mechanisms.

## Conclusions

Together, our findings support a population-level account of temporal order memory in macaque mPPC. Temporally structured population activity during encoding provides a contextual scaffold for representing elapsed time, while coordinated population dynamics and reinstatement-like similarity during retrieval support memory decisions. Rather than relying on explicit timestamping from a specific subset of neurons defined by temporal context or TOJ-related modulation alone, temporal order judgments emerge from distributed representations and their coordinated reactivation. These results extend current models of episodic memory beyond hippocampal–prefrontal frameworks [8,10,16] and position the medial posterior parietal cortex as a key hub linking temporal context encoding to retrieval-related processes [17–19,40,50,66].

## Star* methods

### Resource availability

**Lead contact.** Further information and requests for resources should be directed to and will be fulfilled by the lead contact, Sze Chai Kwok (sze-chai.kwok@st-hughs.oxon.org).

### Experimental model and subject details

**Subjects.** Three male rhesus macaques (*Macaca mulatta*) took part in this study. Two of them (5 and 5.5 years old, weighing 8.3 kg and 8.6 kg) participated in the main experiment and a third one (10 years old and 9.5 kg) participated in Experiment 2. The monkeys were housed in pairs but were singly housed during the period of this study. They were kept on a 12:12 (7:00 AM/7:00 PM) light-dark circle and kept within the temperature range of 18–23 ℃ and humidity between 60% and 80%. The animals were fed twice a day with each portion of at least 180 g monkey chow and pieces of apple (8:30 AM/4:00 PM). Their water supply was only restricted during recording days. No food restriction was imposed throughout the study. The three monkeys were trained on a similar temporal order judgment task prior to this study [29,67]. This study was conducted at the Nonhuman Primate Research Center at East China Normal University. All

animal care, experimental, and surgical procedures were approved by the Institutional Animal Care and Use Committee (permission code: M020150902 and M020150902-2018) at East China Normal University.

**Experimental apparatus.** For all recording sessions, the monkeys, head-restrained, sat in a custom-manufactured Plexiglas monkey chair (29.4 × 30.8 × 55 cm) inside a lead shielded chamber with ventilation during testing. In the main experiment, the monkeys responded with their right hand by touching the stimuli on a 19-inch touch-sensitive screen mounted on a stainless-steel platform. The monkeys were placed within their arm length to the touchscreen. In Experiment 2, the monkey responded with its eye gaze by fixating for at least 1 s on a designated response area on a 31.5-inch LCD monitor (dimension = 714 × 428 mm; resolution = 1,707 × 960). Monkeys' eyes were about 60 and 62 cm away from the screen's top edge and bottom edge. In both experiments, small amounts of water reward were delivered by a liquid dispenser according to a reward contingency described below.

**Surgical procedure.** Anatomical T1 magnetic resonance images were acquired prior to the surgeries to guide us in the placement of the recording chambers. Detailed surgical procedures are reported in detail previously [21]. In brief, the 32-channel recording chambers were placed over the medial posterior parietal cortex for monkey Jupiter (anteroposterior (AP) = − 16.4 mm, mediolateral ML = 5.8 mm; 28° to the right and 14° to the posterior of the transverse plane), monkey Mercury (AP = − 15.4 mm, ML = 7.5 mm; 25° to the right and 9.1° to the posterior), and monkey Mars (AP = − 16.9 mm, ML = −5.1 mm; 7° to the left and 48° to the posterior). Anatomical MRI images with a model of recording electrodes are shown in Fig 1B (upper panel). To confirm the electrode locations, we also acquired CT images (FOV = 8 cm, Voltage = 110 kV, Current = 0.1 mA) for each monkey and aligned the CT image with the MRI image to confirm our recorded sites at the end of the experiments (Fig 1B bottom for Jupiter and Mercury and 1C for Mars).

## Method details (S3 TABLE)

### Temporal order judgment task

In the main experiment, two monkeys were trained to perform a temporal order judgment task based on naturalistic videos. In each trial, the monkey initiated a trial by pressing a colored rectangle in the center of the screen (0.3 ml water). Following a blank screen (range from 1 to 3 s), an 8-s video was presented. After a short period of delay (no delay or 3.6 s delay), two frames extracted from the video were displayed bilaterally on the screen for TOJ. The monkey was required to respond by touching the computer screen, and to choose the frame that was shown earlier in the video to get a reward (1.2 ml water), and the target frame remained alone for 4 s as positive feedback. If the monkey made an error, the screen would be blanked for 4 s with no reward. The temporal distances between the two frames were fixed in all trials (85 frames). To control for temporal similarity [68], we included two delayed conditions. In the immediate condition, the two probe frames were extracted from the first half of the video (i.e., 5th versus 90th frame), with zero retention delay between encoding and retrieval. In the delayed condition, the two probe frames were extracted from the second half of the video (i.e., 95th versus 180th frame), with a retention delay of 3.6 s.

Note that TOJ probe frames were drawn from fixed temporal positions within each video. This design was chosen to equate temporal distance across conditions and thereby isolate neural processes related to temporal order judgment rather than variability in temporal separation or temporal similarity [69]. Prior behavioral work using the same task structure demonstrated that macaques rely on a forward replay–like strategy rather than rote responding [29]. Moreover, the use of naturalistic videos with highly variable content across trials makes frame-counting strategies impractical, and the minimal temporal separation between some probe frames (90th frame as foil versus 95th frame as target, i.e., opposite to the learned TOJ rule) further discourages fixed-location heuristics. These considerations support the interpretation that animals relied on temporal order memory rather than rote frame selection.

In the first stage, we completed 24 sessions of Jupiter and 3 sessions of Mercury. Thirty different videos were used in each block, and each video was shown once corresponding to two different conditions (i.e., immediate versus delayed conditions). Each session contained 6–10 blocks (Jupiter: 8.417 ± 1.100, Mercury: 7.333 ± 1.528) depending on the

monkey's performance. In the second stage, we completed 18 sessions for both Jupiter and Mercury. Fifteen different videos were used twice with each video shown once for immediate and delayed conditions in each block. Each session contained 3–10 blocks (Jupiter: 8.444 ± 1.723, Mercury: 5.667 ± 1.138). In both stages, the same set of videos was randomized across different blocks and reused across three consecutive days. The difference between the stages was the number of videos used per session and the data from both stages were combined for analysis.

In Experiment 2, an additional monkey was tested on the same paradigm. Each session contained 6 blocks, totaling 180 trials per day. The procedure was the same as the main experiment's second stage except for two differences. Firstly, the monkey now needed to make their memory responses by fixating their gaze for at least 1 s on one of two choice boxes (80 × 80 pixels) shown below the TOJ images. Second, the videos used in Experiment 2 consisted of two 4-s clips rather than one single 8-s video. The frames extracted for the temporal order judgment task followed the same rule as the main experiment. Combined with simultaneous eye-tracking and regression analyses of saccades, fixations, and scan-path metrics, this design allowed us to assess whether TOJ-related neural activity could be explained by oculomotor behavior. Accordingly, Experiment 2 is interpreted as a control analysis to rule out eye-movement–related confounds, rather than as an independent test of the main hypotheses.

### Eye-movement recording

An infrared EyeLink 1,000 Plus acquisition device (SR Research) was used to track eye positions at a sampling rate of 1,000 Hz. The illuminator module and the camera were positioned above the monkey's head. An angled infrared mirror was used to capture and re-coordinate monkeys' eye positions. The monkey's right eye was tracked throughout the whole experiment session.

### Electrophysiology recording and spike sorting

On all three monkeys, we used 32-channel semi-chronic Microdrive systems (SC-32, GrayMatter Research) with 1.5 mm inter-electrode spacing for the recording. The headstage of the multichannel utility was connected to an acquisition system (SmartBox, NeuroNexus Technologies, USA) via an amplifier Intan adapter (RHD2000, Intan Technologies) with 32 unipolar inputs. The microelectrode impedance of each channel was in the range of 0.5–2.5 MΩ and measured at the beginning of the session. Spike waveforms above a set threshold were identified with a 1,000 Hz online high-pass filter. Electrophysiological data collection was bandpass filtered from 0.1 to 5,500 Hz and digitized at 30 kHz.

## Quantification and statistical analysis

### Behavioral data analysis

Trials with reaction time longer than 10 s were excluded from analyses (1.71% in main experiment and 3.04% in Experiment 2). RT were analyzed using a linear mixed-effects model with RT as the dependent variable, delay condition (immediate versus delayed) and trial outcome (correct versus error) as fixed effects, and monkey identity as a random effect: *RT ~ Delay Condition + Correctness + (1|Monkey).*

### Eye-movement analysis

Raw eye-movement data was converted from .edf format to .asc format. Saccades were identified by the Eye-Link 1,000 Plus acquisition system with an "SACC" marker during recording. The duration of a saccadic event was defined as the time elapsed starting from when the eye velocity exceeds 15° s$^{-1}$ until when it slows down to below this velocity. We calculated the x and y coordinates for the eye position throughout the video and during the TOJ stage. Scan paths are then created by mapping the coordinates onto the video or TOJ images to produce eye position trajectories. The coordinates during blinks were filled with linear interpolation by using the coordinates 100 ms

before and after a blink. For each trial, the time course of eye-movement data was aligned with the neuronal data for the eye-spike analyses.

## Spike detection and sorting

The raw signal was filtered with a low-cut digital filter (4-pole Butterworth filter, 250 Hz), and we set 3 standard deviations as the threshold of detecting spikes. After removing high amplitude artifacts, we sorted spikes with the Standard E-M algorithm in Offline Sorter (Plexon). We used several criteria to quantify spike sorting quality and to select well-isolated single units, including signal-to-noise ratio, L-ratio, and isolation distance [70] (S1 Fig). We treated clusters identified from the same electrodes on different days as separate neurons.

## Poisson generalized linear models for TOJ-related neural activity

Each neuron was modeled independently using a Poisson GLM with a log link function (MATLAB, stepwiseglm) to identify task-related modulation while controlling behavioral and event-related factors. Candidate regressors included block number within session, reaction time, trial outcome (correct versus error), task condition (immediate versus delayed), and response side (left versus right). Binary event-period regressors indicated activity during pretrial fixation, encoding, delay, TOJ period, feedback (reward versus blank), and inter-trial interval (ITI).

To dissociate sensory-driven, mnemonic, and motor-related components of TOJ, the TOJ period was further decomposed into three nonoverlapping regressors: an early visual-onset window (0–200 ms after probe onset), a post-onset retrieval/comparison window (from 200 ms after probe onset until 200 ms before the behavioral response), and a response-locked motor window (−200 ms to response execution). All event regressors were coded as binary indicators, and firing rates were modeled as the mean firing rate during the corresponding epoch. This decomposition allowed us to dissociate transient stimulus-driven and motor-related responses from later retrieval- and decision-related activity (see Results).

For each neuron, spike counts were modeled independently using a Poisson generalized linear model with a log link function:

$$y_t \sim \text{Poisson}(\lambda_t)$$

$$\begin{aligned}
\log(\lambda_t) = {} & \beta_0 + \beta_{\text{block}}\,\text{Block}_t + \beta_{\text{RT}}\,\text{RT}_t + \beta_{\text{corr}}\,\text{Correctness}_t + \beta_{\text{cond}}\,\text{Condition}_t + \beta_{\text{side}}\,\text{TouchSide}_t \\
& + \beta_{\text{fix}}\,\text{Fixation}_t + \beta_{\text{enc}}\,\text{Encoding}_t + \beta_{\text{delay}}\,\text{Delay}_t + \beta_{\text{fb}}\,\text{Feedback}_t \\
& + \beta_{\text{ITI}}\,\text{ITI}_t + \beta_{\text{TOJvis}}\,\text{TOJ}_{\text{Vis},t} + \beta_{\text{TOJmem}}\,\text{TOJ}_{\text{Mem},t} \\
& + \beta_{\text{TOJmot}}\,\text{TOJ}_{\text{Mot},t}
\end{aligned}$$

where $y_t$ is the spike count in time bin $t$ and $\lambda_t$ is the expected firing rate.

The TOJ regressors correspond to:

$$\text{TOJ}_{\text{Vis}}\ (0-200\ \text{ms after probe onset};\ \text{sensory}-\text{driven}),$$

$$\text{TOJ}_{\text{Mem}}\ (\text{retrieval/comparison period}),\ \text{and}$$

$$\text{TOJ}_{\text{Mot}}\ (\text{motor execution related period}).$$

Model terms were selected using a stepwise procedure based on adjusted pseudo-R², with regressors excluded if adjusted pseudo-R² < 0.005 and retained if > 0.01 [41,48]. Neurons were classified as TOJ cells if at least one TOJ-related regressor was retained as a statistically significant predictor in the final model. Pseudo-R² values are likelihood-based goodness-of-fit measures for Poisson GLMs and do not represent variance explained; classification of TOJ cells was based on statistical significance of TOJ regressors rather than pseudo-R² magnitude.

**Control GLM Incorporating Eye-Movement Parameters.** To rule out eye-movement confounds, we performed a control GLM using data from Experiment 2 that included all regressors from the main model plus eye-movement parameters measured during the TOJ period: saccade frequency, fixation frequency, total saccade path length, and total fixation duration. The same stepwise selection criteria were applied. In this analysis, TOJ cells were more stringently defined as neurons showing significant TOJ-related modulation without any eye-movement regressors retained in the final model.

## Spike-train synchrony analysis using SPIKE-distance

Spike-train synchrony was quantified using the SPIKE-distance metric, a time-resolved, parameter-free measure of spike-train dissimilarity that captures relative spike timing independently of firing rate [47]. For each pair of spike trains recorded within the same session, the SPIKE-distance is computed by evaluating, at each moment in time, the temporal differences between spikes in one train and the nearest preceding and following spikes in the other train. These spike timing differences are interpolated between successive spikes, yielding a continuous measure of instantaneous dissimilarity. Critically, SPIKE-distance is locally normalized by the instantaneous firing rates of the two spike trains, ensuring that synchrony estimates are not confounded by differences in average firing rate. For each pair of neurons, this procedure produces a time-resolved SPIKE-profile representing their moment-by-moment dissimilarity. For each session, pairwise SPIKE-profiles were averaged across all neuron pairs to obtain a population-level SPIKE-distance time course. Pairwise SPIKE-distance values were also summarized in an $n \times n$ dissimilarity matrix, where $n$ denotes the number of simultaneously recorded neurons.

We ran these statistical analyses using a nonparametric permutation-based approach. Specifically, statistical significance of group differences was assessed using 1,000 random permutations. In each permutation, the entire dataset was randomly shuffled and reassigned into two groups while preserving the original group sizes, thereby eliminating any systematic group structure. Independent-samples $t$-statistics were computed for each shuffled dataset to generate a null distribution of $t$-values under the assumption of no true group differences. The significance of the observed $t$-statistics was evaluated using a two-tailed permutation test, with the $p$-value defined as the proportion of permuted $t$-values whose absolute values equaled or exceeded that of the observed statistics. To correct for multiple comparisons in the SPIKE-distance analyses, all $p$-values were further adjusted using the false discovery rate (FDR) procedure (MATLAB *mafdr* function; corrected threshold $p < 0.01$). All SPIKE-distance computations were performed using the SPIKY toolbox. Lower SPIKE-distance values indicate greater spike timing synchrony, whereas higher values reflect greater temporal dissimilarity and lower spike timing synchrony between spike trains.

## Identification of temporal context cells during encoding

We analyzed the spike data of the monkeys via a maximum likelihood estimation script run in Python 3.9. In order to determine whether a neuron had a time-locked response to the onset of video clips, we calculated model fits of nested models for each neuron across all trials considering the time from the onset of image presentation to 8 s after image presentation. The nested models contain three models.

1) The constant model, $F_{const}(t; a_0) = a_0$, which is only determined by a single parameter $a_0$ that predicted the average firing rate at each time $t$.

2) The Gaussian model, $M_{gauss}(t; a_0, a_1, \mu, \sigma) = a_0 + a_1 \frac{1}{\sqrt{2\pi}\sigma} e^{\frac{-(t-\mu)^2}{2\sigma^2}}$.

3) The ex-Gaussian model, $M_{ex-gauss}(t; a_0, a_1, \sigma, \mu, \tau) = a_0 + a_1 \int_{-\infty}^{\infty} e^{-\frac{(t-\mu)^2}{2\sigma^2}} e^{-\frac{t}{\tau}} dt$, which is represented by the convolution of a Gaussian function with an exponentially decaying function.

Since the ex-Gaussian model degrades into either a Gaussian function (as $\tau \to 0$) or an exponential function starting at $\mu$ (as $\sigma \to 0$), this model performs well in describing both temporal context cells and time cells. As such, neurons which are better fitted by the ex-Gaussian model are considered responsive. We selected neurons with three criteria: 1) were better fitted by the ex-Gaussian model at the 0.05 level, 2) changed their firing rate by at least 2 Hz, and 3) reached a firing rate of at least 4 Hz. Fits of nested models for each neuron are analyzed via a likelihood ratio test.

## Population temporal decoding using linear discriminant analysis

Temporal information was decoded from population activity using a LDA classifier with k-fold cross-validation to ensure robust estimation and to avoid overfitting. For each trial, the 8-s encoding period was divided into 32 nonoverlapping 250-ms time bins. Population activity in each bin was represented as a vector of firing rates across all neurons recorded in that session. Temporal decoding was performed using a LDA classifier implemented in MATLAB. Rather than a single odd–even split, we employed a 5-fold cross-validation procedure. For each session, trials were randomly partitioned into five folds using a stratified approach, and each trial served as test data exactly once. In each iteration, the classifier was trained on four folds (80% of trials) and tested on the remaining fold (20% of trials), such that each trial served as test data exactly once. To ensure equal contribution of neurons and avoid duplication, trials were subsampled to the minimum trial count within each monkey (Jupiter: 180; Mercury: 90; Mars: 120), without resampling with replacement. This procedure yielded five independent estimates of decoding performance, which were averaged to obtain session-level results. Decoding performance was quantified as mean absolute temporal decoding error. Statistical significance was assessed using permutation tests (1,000 iterations), with shuffled time bin labels generating a null distribution. Performance was considered above chance if fewer than 10 permutations yielded lower error ($p < 0.01$).

**Robustness of temporal decoding to early-bin exclusion.** To ensure that temporal decoding performance was not driven disproportionately by early encoding responses, we repeated the LDA analysis after progressively excluding early portions of the video. Specifically, time bins corresponding to the first 1–6 s of the encoding period were removed in 1-s increments, and decoding was recomputed using the remaining bins. For each truncation level, decoding accuracy was compared against a permutation-based null distribution generated by shuffling time bin labels (1,000 permutations).

## Representational similarity analysis

To reveal how temporal context drifts over time during encoding, neural spike data were binned into 250 ms windows across the encoding period (8 s in total) and z-score normalized to control baseline firing rate differences across neurons. We computed Pearson correlation coefficients between all pairs of time bins to construct representational dissimilarity matrices (RDMs), where the correlation strength between temporally distant bins reflects the stability of neural population states over time.

## Encoding–retrieval similarity by Mahalanobis distance

To quantify the similarity between population activity patterns during encoding and retrieval, we computed trial-wise Mahalanobis distances between population firing rate vectors. For each session and trial, we constructed population vectors consisting of the mean firing rates of all simultaneously recorded neurons during the video encoding period and the TOJ period, respectively. The squared Mahalanobis distance between these vectors was calculated using MATLAB 'mahal' function, providing a multivariate measure of dissimilarity that accounts for covariance structure across neurons.

 

Because the number of recorded neurons varied across sessions, Mahalanobis distances were normalized by twice the number of neurons in each session to yield a comparable distance index across sessions. Lower Mahalanobis distance values indicate greater similarity between encoding and retrieval population states, consistent with trial-wise reinstatement of temporal context during temporal order judgment.

**Permutation control for encoding–retrieval similarity.** To assess whether encoding–retrieval similarity exceeded chance levels, we additionally performed permutation controls in which encoding and retrieval population matrices were independently shuffled across trials and recomputed distances. This procedure preserved marginal firing rate statistics while disrupting trial-specific correspondences, allowing us to test whether observed similarities reflected meaningful reinstatement rather than nonspecific visual or firing rate effects.

**Control analysis for early visual encoding confounds.** To rule out the possibility that encoding–retrieval similarity effects were driven by stronger visual responses early in the video, we conducted a linear mixed-effects analysis on trial-wise Mahalanobis distance values. The model included trial correctness (correct versus incorrect), video half (first versus second half of the video during encoding), and delay condition (immediate versus delayed) as fixed effects, along with their interactions, and monkey identity as a random effect. The model was specified as:

$$\text{MDist}_{ij} = \beta_0 + \beta_1 \text{Correctness}_{ij} + \beta_2 \text{VideoHalf}_{ij} + \beta_3 \text{Delay}_{ij} + \beta_4 (\text{Correctness} \times \text{VideoHalf})_{ij} + (1 \mid \text{Monkey}_j)$$

This analysis allowed us to test whether encoding–retrieval similarity depended on whether the Mahalanobis distance was computed based on the early or late portion of the video, independent of memory accuracy.

## Supporting information

**S1 Data. Data for** Figs 1–6 **and** S1–S3.
(XLSX)

**S1 Fig. Spike sorting quality metrics.** (A) Distribution of mean firing rates for all recorded single units across three monkeys (401 and 275 cells from the main experiment; 198 cells from Experiment 2). The population mean firing rate was $12.29 \pm 14.38$ spikes/s (mean $\pm$ SD). (B) Histogram of signal-to-noise ratios (SNRs) computed from the mean waveform of each neuron (mean $\pm$ SD: $2.50 \pm 0.73$). (C and D) Distributions of L-ratio values calculated from 2D (C) and 3D (D) feature spaces (2D: $0.59 \pm 1.56$; 3D: $0.39 \pm 0.63$). (E and F) Distributions of isolation distance computed from 2D (E) and 3D (F) feature spaces (2D: $6.20 \pm 10.76$; 3D: $7.12 \pm 11.49$). Together, these metrics indicate that recorded units were well-isolated and met standard criteria for single-neuron quality across experiments. It is associated with main Fig 1. Data underlying this figure is available in S1 Data.
(TIF)

**S2 Fig. Robustness of temporal decoding to removal of early encoding bins.** To ensure that temporal decoding was not driven by stronger neural responses early in the encoding period, we repeated the linear discriminant analysis after progressively excluding early time bins from the video in 1-s increments. For each exclusion level, posterior probabilities over decoded time bins and corresponding decoding errors were computed for each monkey. Decoder performance remained significantly better than chance (permutation test, $p < 0.01$) even after removal of up to 6 s of early encoding activity, indicating that population-level temporal information was distributed across the entire video and not dominated by early visual responses. Posterior probability matrices and corresponding decoding performance are shown for each monkey (A and B: Jupiter, C and D: Mercury, E and F: Mars). For each monkey, the top left panel shows decoding results with the first 1 s removed, with subsequent panels corresponding to removal of the first 2–5 s. Panels B, D, and F show the corresponding decoder errors. The decoded time bin with the highest posterior probability is marked by a red dot, and 1-s

intervals are indicated by red arrows. Dashed line marked by '**' denote significance at $p < 0.01$. It is associated with main Fig 5. Data underlying this figure is available in S1 Data.
(TIF)

**S3 Fig. Eye-movement control analysis: example neurons.** (A and B) Normalized firing rates of six example neurons are shown in relation to eye position during the TOJ period for three correct trials (A) and three incorrect trials (B). For each trial, eye gaze fixations are shown in the upper panels and saccadic scan paths in the lower panels. Neuronal firing rates are overlaid on fixation locations (colored discs) and along scan paths (colored dotted trajectories), with color indicating normalized firing rate (see color bars). The beginning and end of each trial are marked as "Start" and "End", and arrows indicate saccade direction and amplitude. These examples illustrate that TOJ-related neural activity cannot be trivially explained by fixation location or scan-path, consistent with the GLM-based eye-movement control analyses reported in the main text. All still frames displayed here were generated and assembled by the authors for illustrative purposes and do not contain any third-party copyrighted material. See also S1 Video. Data underlying this figure is available in S1 Data.
(TIF)

**S1 Table. Distribution of temporal context (TC) cells and their overlap with TOJ cells across monkeys.** Notes: TC cells were defined based on ex-Gaussian model fits during the video encoding period (see Methods). Percentages indicate proportions relative to the indicated denominator.
(XLSX)

**S2 Table. Identification of temporal order judgment (TOJ) cells using main and eye-movement–controlled GLMs.** Notes: For Experiment 2 (Monkey Mars), TOJ cells were re-identified using a control Poisson GLM including eye-movement regressors (saccade frequency, fixation frequency, scan-path length, and fixation duration). Percentages are relative to total cells recorded for each monkey.
(XLSX)

**S3 Table. Key resources used in this study.** Summary of experimental models, software, and hardware. The table lists each resource, its source, and relevant identifiers (e.g., model numbers or URLs) to facilitate reproducibility.
(DOCX)

**S1 Video. Reconstructed example trial with eye-movement tracking.** This movie illustrates the structure of a temporal order judgment trial, including fixation, video encoding, delay, decision, and feedback. The green dot indicates the monkey's eye gaze. The visual content has been reconstructed for illustrative purposes and does not include the original experimental stimuli. It does not contain any third-party copyrighted material.
(MP4)

## Acknowledgments

We would like to thank Edmund Rolls for helping the team with NHP electrophysiology and dedicate this work to late Yong-di Zhou, who founded the Nonhuman Primate Research Center at East China Normal University. SZ received support from the Kobayashi Foundation.

## Author contributions

**Conceptualization:** Sze Chai Kwok.

**Data curation:** Shuzhen Zuo, Zhiyong Jin.

**Formal analysis:** Shuzhen Zuo, Chenyu Wang.

**Funding acquisition:** Sze Chai Kwok.

**Investigation:** Shuzhen Zuo, Zhiyong Jin, Xufeng Zhou, Ning Su, Jianhua Liu, Makoto Kusunoki, Sze Chai Kwok.

**Methodology:** Shuzhen Zuo, Lei Wang, Zhiyong Jin, Thomas J. McHugh, Makoto Kusunoki, Sze Chai Kwok.

**Project administration:** Sze Chai Kwok.

**Resources:** Sze Chai Kwok.

**Software:** Shuzhen Zuo.

**Supervision:** Sze Chai Kwok.

**Validation:** Shuzhen Zuo.

**Visualization:** Shuzhen Zuo, Chenyu Wang.

**Writing – original draft:** Shuzhen Zuo, Sze Chai Kwok.

**Writing – review & editing:** Shuzhen Zuo, Sze Chai Kwok.

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
