## [Editor Report · Decision Letter 0]

12 Feb 2026

Dear Sze Chai,

Thank you for submitting your manuscript entitled "Neural signatures for temporal order memory in the macaque medial posterior parietal cortex" for consideration as a Research Article by PLOS Biology and apologies for the delay in getting back to you.

Unfortunately, I was not able to receive advice from the academic editor on your revision. However, after discussing your submission with the team, I am writing to let you know that we would like to send your submission back for external peer review.

Once your full submission is complete, your paper will undergo a series of checks in preparation for peer review. After your manuscript has passed the checks it will be sent out for review. To provide the metadata for your submission, please Login to Editorial Manager (https://www.editorialmanager.com/pbiology) within two working days, i.e. by Feb 14 2026 11:59PM.

Kind regards,

Christian

Christian Schnell, PhD

Senior Editor

PLOS Biology

cschnell@plos.org

---

## [Decision Letter · Decision Letter 1]

17 Mar 2026

Dear Sze Chai,

Thank you for your patience while we considered your revised manuscript "Neural signatures for temporal order memory in the macaque medial posterior parietal cortex" for consideration as a Research Article at PLOS Biology. Your revised study has now been evaluated by the PLOS Biology editors, the Academic Editor and two of the original reviewers.

In light of the reviews, which you will find at the end of this email, we are pleased to offer you the opportunity to address the remaining points from Reviewer 3 in a revision that we anticipate should not take you very long. We will then assess your revised manuscript and your response to the reviewers' comments with our Academic Editor aiming to avoid further rounds of peer-review, although we might need to consult with the reviewers, depending on the nature of the revisions.

**IMPORTANT - SUBMITTING YOUR REVISION**

*Resubmission Checklist*

*Published Peer Review*

*PLOS Data Policy*

*Blot and Gel Data Policy*

Sincerely,

Christian

Christian Schnell, PhD

Senior Editor

PLOS Biology

cschnell@plos.org

REVIEWS:

Reviewer #1: The authors have sufficiently addressed all my concerns, and I recommend the paper for publication.

Reviewer #3 (Liping Wang has signed his report): In this revision, the authors went to great lengths to address my concerns, and adequately so for the most part. While I appreciate the substantial effort evident from the overhaul of the manuscript, I do still have a few lingering issues that would hopefully improve the article further if properly addressed.

1. Regarding major issue #6 in our original review, since the authors have revised the results in Figure 1E to focus on comparing reaction time (RT) differences between the immediate and delayed conditions, I suggest they accordingly update the presentation of Figure 1E. Specifically, they should clearly label the groups being compared (The key comparison should be Frame 5 vs. Frame 90, right?) and include the corresponding statistical results to facilitate reader understanding. Furthermore, since the 90- and 180-frame trials represent error trials and the primary objective appears to be comparing immediate vs. delayed conditions, it seems unclear why these error trials were included in this analysis. Notably, while the figure displays results for the two monkeys separately, the main text appears to report statistical results based on pooled data from both animals. Does the conclusion stated in the main text also hold true for monkey Mercury? Generally speaking, I recommend that the authors double-check their revisions to ensure consistency between the figure(s) and the main text.

2. While the video time decoding and encoding-retrieval similarity analyses yielded encouraging results, they appear only tangentially related to previous classification efforts that identified populations of TOJ/temporal context cells. One is left wondering, for instance, whether temporal context cells seen in Figure 4 preferentially contributed to decoding in Figure 5, or how distance metrics in Figure 6 would be impacted when considering only temporal information-carrier neurons versus others. This disjunction, alongside newly introduced sidetracking sections that addressed various issues brought up by reviewers, made reading through the manuscript a fairly fragmentary experience. Perhaps the authors should consider shoring up their main narrative throughout the Results section, so that the readers could follow along a logical chain of events that would bridge the gap between cell type identification and task information representation.

3. Figure 2A, the text of the ticks on the x-axis is overlapping with the axis line. The bottom edge of Figure 5G was cropped off.

---

## [Editor Report · Decision Letter 2]

25 Mar 2026

Dear Sze Chai,

Thank you for your patience while we considered your revised manuscript "Neural signatures for temporal order memory in the macaque medial posterior parietal cortex" for publication as a Research Article at PLOS Biology. This revised version of your manuscript has been evaluated by the PLOS Biology editors and the Academic Editor.

Based on our Academic Editor's assessment of your revision, we are likely to accept this manuscript for publication, provided you satisfactorily address the following data and other policy-related requests:

* We would like to suggest a different title to improve its accessibility for our broad audience:

Temporal context cells in the macaque medial parietal cortex integrate temporally extended experience with memory-guided decisions

* Please add the links to the funding agencies in the Financial Disclosure statement in the manuscript details.

* DATA POLICY:

Regardless of the method selected, please ensure that you provide the individual numerical values that underlie the summary data displayed in the following figure panels as they are essential for readers to assess your analysis and to reproduce it: 1E, 2ABDEF and 6B.

* CODE POLICY

Per journal policy, if you have generated any custom code during the course of this investigation, please make it available without restrictions. Please ensure that the code is sufficiently well documented and reusable, and that your Data Statement in the Editorial Manager submission system accurately describes where your code can be found. More information on our Code Policy, what and how to share can be found here: https://journals.plos.org/plosbiology/s/code-availability

* We do not have a word count limit. Please move the supplementary text to the main manuscript, so readers can easily access this information.

We expect to receive your revised manuscript within two weeks.

*Published Peer Review History*

*Press*

Sincerely,

Christian

Christian Schnell, PhD

Senior Editor

cschnell@plos.org

PLOS Biology

---

## [Editor Report · Decision Letter 3]

1 Apr 2026

Dear Sze Chai,

Thank you for the submission of your revised Research Article "Neural population dynamics and temporal context cells in macaque medial parietal cortex support temporal order memory" for publication in PLOS Biology. On behalf of my colleagues and the Academic Editor, Mathew Diamond, I am pleased to say that we can in principle accept your manuscript for publication, provided you address any remaining formatting and reporting issues. These will be detailed in an email you should receive within 2-3 business days from our colleagues in the journal operations team; no action is required from you until then. Please note that we will not be able to formally accept your manuscript and schedule it for publication until you have completed any requested changes.

PRESS

We frequently collaborate with press offices. If your institution or institutions have a press office, please notify them about your upcoming paper at this point, to enable them to help maximize its impact. If the press office is planning to promote your findings, we would be grateful if they could coordinate with biologypress@plos.org. If you have previously opted in to the early version process, we ask that you notify us immediately of any press plans so that we may opt out on your behalf.

Sincerely,

Christian

Christian Schnell, PhD

Senior Editor

PLOS Biology

cschnell@plos.org